# Sound Absorption Properties of Perforated Recycled Polyurethane Foams Reinforced with Woven Fabric

**DOI:** 10.3390/polym12020401

**Published:** 2020-02-10

**Authors:** Roberto Atiénzar-Navarro, Romina del Rey, Jesús Alba, Víctor J. Sánchez-Morcillo, Rubén Picó

**Affiliations:** 1Instituto de Investigación para la Gestión Integrada de Zonas Costeras, Universitat Politècnica de València, 46730 València, Spain; victorsm@fis.upv.es (V.J.S.-M.); rpico@fis.upv.es (R.P.); 2Centro de Tecnologías Físicas: Acústica, Materiales y Astrofísica, Universitat Politècnica de València, 46730 València, Spain; roderey@fis.upv.es (R.d.R.); jesalba@fis.upv.es (J.A.)

**Keywords:** sound absorption, textile fabrics, recycled polyurethane foam, finite element

## Abstract

The acoustic properties of recycled polyurethane foams are well known. Such foams are used as a part of acoustic solutions in different fields such as building or transport. This paper aims to seek improvements in the sound absorption of these recycled foams when they are combined with fabrics. For this aim, foams have been drilled with cylindrical perforations, and also combined with different fabrics. The effect on the sound absorption is evaluated based on the following key parameters: perforation rate (5% and 20%), aperture size (4 mm and 6 mm), and a complete perforation depth. Experimental measurements were performed by using an impedance tube for the characterization of its acoustic behavior. Sound absorption of perforated samples is also studied—numerically by finite element simulations, where the viscothermal losses were considered; and analytically by using models for the perforated foam and the fabric. Two textile fabrics were used in combination with perforated polyurethane samples. Results evidence a modification of the sound absorption at mid frequencies employing fabrics that have a membrane-type acoustic response.

## 1. Introduction

Nowadays, society demands a greater sensitivity towards compatibility between progress and respect for the environment. Favoring the regeneration and sustainability of the ecosystem of the territory is only possible with proper management and responsible and efficient use of natural resources. Researchers are focusing on this idea as written in the Horizon2020 Challenge 5 and by the Sustainable Development Goals (SDGs). In particular, the SDG 12 of the new European program called Horizon2030, indicates that the use of sustainable raw materials with a lower carbon footprint and secondary materials are crucial. These green recycled materials are compatible from the environmental point of view and may be an alternative to address noise and vibration issues.

According to the World Health Organization (WHO) [1], noise pollution is now gaining global attention because it has become a major world health problem affecting everyday life of people. Exposure to excessive noise may cause undesired physiological and psychological effects on humans [2].

Some adverse effects on human health caused by the long-term exposure to high levels of unwanted sound are hearing loss, stress, fatigue, insomnia, high blood pressure, or serious cardiovascular disorders such as heart attacks. Hence, there is a need to investigate the control of noise pollution effectively.

Numerous previous studies can be found in the literature concerning the use of materials obtained from recycled products for noise mitigation. A sustainable use of natural resources is pursued by developing new materials using waste as raw material, reusing them or recycling them in part or in its whole. Eco-materials with absorbent acoustic properties—like recycled plastic bottles (PET), natural fibers such as kenaf [3], coconut [4], sheep wool [5], or cotton [6]—have been proposed. Other porous materials obtained from the recycling of textile waste are cellular materials (foams), like recycled polyurethane (RPU) [7], that also show interesting acoustic properties [8]. They can be considered as an alternative to classical materials.

RPU is an eco-friendly polymer obtained from manufacturing processes of the textile industry and it is widely used as sound absorbing material due to its high efficiency on sound absorption at mid and high frequencies. In [9], how to predict the acoustic behavior of RPU samples by using two different empirical models is presented. More recently, a novel material was presented [10] based on the combination of polyurethane foam and bamboo leaves particles by using different contents with the aim to improve acoustic properties.

Theoretical models are used to describe the acoustic propagation and absorption in different materials [11,12,13]. In 1994, Voronina presented in [14] an empirical model for homogeneous materials that takes into account the structural characteristic of fibrous material. In 2005, Umnova et al. proposed in [15] an acoustic method in order to estimate the porosity and tortuosity of rigid-porous materials.

The perforation of foams can change and improve the sound absorption. Among the contributions of the last decade, Zhang et al. investigated in [16] the influence of the pore cell size and the open porosity of RPU samples in their sound absorption coefficient. Tiuc et al. examined in [17] the influence of the perforation on the sound absorption coefficient and they accounted for an increase of the sound absorption coefficient at frequencies above 1200 Hz by using perforated samples.

Drilling technologies are used in other materials as micro-perforated panels (MPP). In this context, Chevillotte et al. investigated in [18] the influence of the microstructural parameters as perforation diameter, pore distribution, and pore diameter on the sound absorption coefficient of the perforated aluminum foam sample. Lou et al. showed in [19] the influence of the aperture size, the perforation rate, perforation depth, and back air-cavity on the flexible perforated composite plate. In both studies, the results revealed that perforation could enhance the sound absorption. Lin et al. [20] studied perforated rigid PU foam plates. Results showed that the sound absorption of perforated plates slightly increased when the perforation rate decreases. Lately, Xia et al. presented their work based on foam macrostructures in [21] where they investigated the effects of different types of perforation and air-cavities on the sound absorption coefficient. The results revealed that the foams with half-hole presented a good sound absorption behavior for frequencies greater than 2500 Hz.

With the goal to describe the effects of perforation with this type of materials, different models have been proposed. In 2001, Atalla et al. presented in [22] a 3-D numerical finite element model in order to predict the absorption coefficient of non-homogeneous porous material. In 2003, Olny and Boutin [23] used the homogenization method for periodic structures (HPS) and the simplified model proposed by Zwikker and Kosten [24] in order to study the wave propagation in a porosity media with two periodic interconnected network of pores. The results showed the effect of some design parameters (meso-porosity, type of lattice, size of the pores) on the sound absorption. In 2005, Sgard et al. analyzed in [25] models for predicting the acoustic wave transmission of double porosity media and they investigated the effect of the aperture size, perforation rate, and air-cavities on the sound absorption coefficient. Results showed a good agreement between the measurements and the models. Recently, Carbajo et al. used in [26] the double porosity media to acoustically model rigid porous materials.

New acoustic solutions using thin fiber membranes in combination with porous materials have been investigated recently in order to modify the acoustic absorption of the system fabric-RPU. In 2012, Ekici et al. presented in [27] a sound absorbent material made of tea-leaf fibers and luffa cylindrica with RPU. The number of tea-leaf fibers is shown to increase the sound absorption coefficient for all frequency ranges. In 2014, del Rey et al. studied in [28] the sound absorption improvements that can be achieved with the combination of nanopaper and different types of base substrates. Results showed that the nanopaper enhances the acoustic absorption properties of fibrous and porous materials at mid and high frequencies. Segura-Alcaraz et al. demonstrated in [29] the influence of microfiber fabric placed on the top of a nonwoven structure. Results showed that the selectivity of the absorption depends on the type of fabric used.

This work investigates the normal incidence sound absorption coefficient in different configurations. These configurations are composed of a layer of thin fabric acting as a membrane on the top of the meso-perforated porous materials. These foams were perforated according to two design parameters (perforation rate and aperture size of the holes). For this study, perforations were carried out using two different technologies: laser and milling machine. Compared to previous works of other authors, this paper focuses on controlling the sound absorption using different textile fabrics in the system membrane—RPU. Results were compared to those obtained numerically and analytically with a good agreement between them.

This paper is organized as follows. Section 2 briefly describes the theoretical background, where different models have been developed in order to study the acoustic properties of both RPU foam and textile fabric. The acoustic properties of porous media are described by the Johnson–Champoux–Allard model [12,13]. The Olny and Boutin [23] model is used for double porosity media. Pieren’s model [30] is used to predict the acoustic behavior of the thin membranes. In Section 3, the numerical model based on finite elements is introduced to validate the theoretical model and the experimental results. The perforated foam fabrication and the experimental setup using the impedance tube are detailed in Section 4. In Section 5, the experimental, theoretical and numerical results are compared and discussed. Finally, as a result of this work, Section 6 includes the most relevant conclusions.

## 2. Theoretical Background

In this work, different theoretical models are used to predict the properties of the different materials under investigation. These models are reviewed in this section.

### 2.1. Johnson–Champoux–Allard Model

Johnson–Champoux–Allard (JCA) proposed in [12,13] a model to describe the acoustic behavior of porous media throughout the frequency range. Here, this model is used to describe the dissipative viscothermal effects of the RPU foam. The expression for the complex dynamic density ($\rho_{\mathrm{JCA}},\;\mathrm{kg}/m^{3}$) includes the visco-inertial effects (Equation (1)) and the expression for the complex bulk modulus ($K_{\mathrm{JCA}},\;\mathrm{Pa}$) includes the thermal effects (Equation (2)). Both equations involve parameters such as airflow resistivity, two characteristic lengths, porosity and tortuosity as
(1)$$\rho_{\mathrm{JCA}}=\frac{\alpha_{\infty }\rho_{0}}{\emptyset }\left(1+\frac{\sigma \emptyset }{{j\omega \rho }_{0}\alpha_{\infty }}\sqrt{1+\frac{4{j\omega \alpha }_{\infty }^{2}{\mu \rho }_{0}}{\sigma^{2}\Lambda^{2}\emptyset^{2}}}\right)$$
(2)$$K_{\mathrm{JCA}}=\frac{{\gamma P}_{0}}{\emptyset }{\left(\gamma -\left(\gamma -1\right){\left(1+\frac{8\mu }{{{j\omega \Lambda }^{\prime }}^{2}N_{\mathrm{Pr}}\rho_{0}}\sqrt{1+\frac{{{j\omega \Lambda }^{\prime }}^{2}N_{\mathrm{Pr}}\rho_{0}}{16\mu }}\right)}^{-1}\right)}^{-1}$$

In Equations (1) and (2), the physical parameters used in this work for the properties of air are the dynamic viscosity μ = 1.84$\;\cdot 10^{-5}\;\mathrm{Pa}\cdot s$, the air density $\rho_{0}$ = $1.213\;\mathrm{kg}\cdot m^{-3}$, the Prandtl number $N_{\mathrm{Pr}}$ = 0.708 at normal atmospheric pressure and temperature, and the speed of sound in air $c_{0}$ = 343 m/s. The macroscopic parameters of the porous media are porosity ∅, tortuosity $\alpha_{\infty }$, airflow resistivity $\sigma \;\left(\mathrm{Pa}\cdot s\cdot m^{-2}\right)$, viscous characteristic length Λ, and thermal characteristic length $\Lambda^{\prime }$ of the porous material are defined in the next section.

#### Estimation of JCA Parameters

Four parameters involved in JCA modelling (porosity, tortuosity, viscous, and thermal characteristic length) have been estimated in order to predict the complex characteristic impedance, Z_c_, and the complex wave number, k. These parameters are necessary to describe the acoustic behavior of the porous media.

Porosity (∅) is defined as the ratio of the volume of holes over the total volume. For homogeneous materials, it can be estimated by using the Voronina model (1994) [14]
(3)$$\emptyset =1-\frac{\rho_{m}}{\rho_{f}}$$
where $\rho_{m}$ is the bulk density of the porous media and $\rho_{f}$ is the density of the fibers.

Tortuosity $\left(\alpha_{\infty }\right)$ describes the influence of the internal structure of porous sound absorbing materials for the prediction of the acoustical properties. According to [15], it relates to porosity as
(4)$$\alpha_{\infty }=1+\left(1-\emptyset \right){\left(2\emptyset \right)}^{-1}$$

Viscous characteristic length (Λ) is a parameter used to describe the dissipative visco-inertial effects [12], which is related to the smallest pore size. Allard and Champoux introduced in [13] a thermal characteristic length ($\Lambda^{\prime }$) to characterize the thermal dissipation effect, which is related to the largest pore size and can be estimated as
(5)$$L=\frac{4\rho_{m}}{{\pi \rho }_{f}d^{2}}$$
(6)$$\Lambda^{\prime }=2\Lambda =\frac{1}{\pi \mathrm{RL}}$$
where L is the total length of the fiber per unit volume of the material, d is the mean fiber diameter, R = d/2, $\rho_{m}$ is the bulk density of the porous medium and $\rho_{f}$ is the density of the fibers.

### 2.2. Olny and Boutin Model

Olny and Boutin method [23] is used when meso-perforated o double porosity materials with two periodic interconnected networks of pores are considered. Homogenization method for periodic structures (HPS) is used to describe the wave propagation in the double porous media from the macroscopic point of view. The complex dynamic density and the complex bulk modulus of the homogeneous medium are given by
(7)$$\rho_{\mathrm{dp}}={\left(\frac{1-\emptyset_{M}}{\rho_{\mathrm{JCA}}}+\frac{1}{\rho_{M}}\right)}^{-1}$$
(8)$$K_{\mathrm{dp}}={\left(\frac{1-\emptyset_{M}}{K_{\mathrm{JCA}}}+\frac{1}{K_{M}}\right)}^{-1}$$
where $\emptyset_{M}$ is the meso-porosity. The scale ratio between the mesopores and the micropores is low ($\approx 10^{-3}$). The complex dynamic density and the complex bulk modulus of the cylindrical holes of circular cross-section can be computed using the simplified model proposed by Zwikker and Kosten [24]
(9)$$\rho_{M}=\frac{\rho_{0}}{\emptyset_{M}\left(1-\frac{2J_{1}\left(s\sqrt{-j}\right)}{s\sqrt{-j}J_{0}\left(s\sqrt{-j}\right)}\right)}$$
(10)$$K_{M}=\frac{{\gamma P}_{0}}{\emptyset_{M\;}\left(\gamma -\left(\gamma -1\right)\left(1-\frac{2J_{1}\left(N_{\mathrm{Pr}}s\sqrt{-j}\right)}{N_{\mathrm{Pr}}s\sqrt{-j}J_{0}\left(N_{\mathrm{Pr}}s\sqrt{-j}\right)}\right)\right)}$$
where s = $R_{1}{\left(\frac{{\omega \rho }_{0}}{2\mu }\right)}^{1/2}$, $R_{1}$ is the radius of the circular cross-section (cylindrical mesopores) and ω is the angular frequency, $J_{0}$ and $J_{1}$ are the Bessel functions of the first kind and zero-th and first order, respectively.

Finally, the frequency-dependent complex characteristic impedance $\left(Z_{c},\mathrm{Pa}\cdot s/m\right)$ and the complex wave number (k,rad/m) expressions can be estimated as
(11)$$Z_{c}=\sqrt{\rho_{\mathrm{dp}}k_{\mathrm{dp}}}$$
(12)$$k=\omega \sqrt{\frac{\rho_{\mathrm{dp}}}{k_{\mathrm{dp}}}}$$

These two magnitudes allow to calculate the surface impedance of a rigidly backed sample of thickness t.
(13)$$Z_{S}=-{\mathrm{jZ}}_{c}\coth\left(\mathrm{kt}\right)$$

Finally, the reflection and absorption coefficients are calculated from Equation 13 as
(14)$$r=\frac{Z_{S}-\rho_{0}c_{0}}{Z_{S}+\rho_{0}c_{0}}$$
(15)$$\alpha_{n}=1-{\left|r\right|}^{2}$$

### 2.3. Pieren’s Model

The Pieren model [30] is used to calculate the impedance of a fabric. In this model, textile fabric can be considered acoustically thin compared to the wavelength of sound. In this work, this condition is fulfilled for both fabrics in the whole working frequency range. The acoustic impedance of the fabric is defined as
(16)$$Z_{\mathrm{tex}}=\frac{{j\omega m}_{f}R_{s}}{{j\omega m}_{f}+R_{s}}=\frac{R_{s}{\left({\omega m}_{f}\right)}^{2}}{R_{s}^{2}+{\left({\omega m}_{f}\right)}^{2}}+j\frac{R_{s}^{2}{\omega m}_{f}}{R_{s}^{2}+{\left({\omega m}_{f}\right)}^{2}}$$
where $m_{f}$ is the surface mass density and $R_{s}$ is the specific airflow resistance.

The surface impedance of the complete absorbent structure $\left(Z_{\mathrm{in}}\right)$ can be characterized by the impedance of the textile fabric ($Z_{\mathrm{tex}})$ plus the impedance of the polyurethane foam ($Z_{S})$, leading to
(17)$$Z_{\mathrm{in}}=Z_{\mathrm{tex}}+Z_{S}=\frac{R_{s}{\left({\omega m}_{f}\right)}^{2}}{R_{s}^{2}+{\left({\omega m}_{f}\right)}^{2}}+j\frac{R_{s}^{2}{\omega m}_{f}}{R_{s}^{2}+{\left({\omega m}_{f}\right)}^{2}}-{\mathrm{jZ}}_{c}\coth\left(\mathrm{kt}\right)$$

### 2.4. Delany and Bazley Model

This semi-empirical model describes the acoustic behavior in terms of several non-intrinsic physical parameters, which allow to minimize propagation errors on the basis of experimentally obtained parameters [11].

This involves obtaining the coefficients C_i_ (1 ≤ i ≤ 8) that best fit the equations presented for the materials under study
(18)$$\alpha =\left(\frac{2\pi f}{c_{0}}\right)\left[C_{5}{\left(\frac{\rho_{0}f}{\sigma }\right)}^{-c_{6}}\right]$$
(19)$$\beta =\left(\frac{2\pi f}{c_{0}}\right)\left[1+C_{7}{\left(\frac{\rho_{0}f}{\sigma }\right)}^{-c_{8}}\right]$$
(20)$$Z_{R}=\rho_{0}c_{0}\left[1+C_{1}{\left(\frac{\rho_{0}f}{\sigma }\right)}^{-c_{2}}\right]$$
(21)$$Z_{I}=-\rho_{0}c_{0}\left[C_{3}{\left(\frac{\rho_{0}f}{\sigma }\right)}^{-c_{4}}\right]$$
where α and β are the real and imaginary part of the propagation constant Γ, $Z_{R}$, and $Z_{I}$ are the real and imaginary part of the characteristic impedance Z, σ is the airflow resistivity in the direction of wave propagation and f is the working frequency range.

From the characteristic impedance Z and the propagation constant Γ, it is possible to obtain the sound absorption coefficient by using Equations (13) and (15).

The experimental data, both the normal incidence sound absorption coefficient ($\alpha_{n})$ and the airflow resistivity (σ) are necessary to obtain the C_i_ parameters of the adjustment. An iterative method based on reducing the function of the quadratic error has been used in order to obtain the coefficients that best describe the acoustic behavior of the samples under study,
(22)$$\varepsilon =\overset{N}{\underset{i=1}{\sum }}{\left(\alpha_{n,i}-{\hat{\alpha }}_{n,i}\right)}^{2}$$
where $\alpha_{n,i}$ represents the normal incidence sound absorption coefficient values, measured for a textile fabric at i-th frequency and ${\hat{\alpha }}_{n,i}$ is the estimated previous value from Equations (18)–(21).

The coefficients follow from the minimization of the error function, i.e., by solving the system
(23)$$\frac{\partial \epsilon }{\partial C_{i}}=2\sum_{i=1}^{N}\left(\alpha_{n,i}-{\hat{\alpha }}_{n,i}\right)\frac{\partial {\hat{\alpha }}_{n,i}}{\partial C_{i}}=0,\;1\leq i\leq 8$$

## 3. Numerical Model for Porous Media

A numerical study using the Finite Element Method (FEM) is also performed to validate the theoretical model and the experimental results. This method solves the Helmholtz equation
(24)$$\nabla^{2}\hat{p}\left(\omega \right)+k^{2}\hat{p}\left(\omega \right)=0$$
in a discretized domain, k is the lossless wave number (k=ω/c), $\nabla^{2}=\frac{\partial^{2}}{\partial x_{1}^{2}}+\frac{\partial^{2}}{\partial x_{2}^{2}}$ represents the Laplacian and $\hat{p}$ is the complex amplitude of the acoustic pressure.

The sound source is modeled by a plane wave of unit amplitude. The frequency range used in the numerical model is the same as in the experimental method, from 400 Hz to 3150 Hz. The mesh size (Δx) is chosen to ensure convergence, that is, Δx
<
$\lambda_{\min}$/8 ≈14 mm.

The physical magnitude analyzed is the total sound pressure field $\left(p_{t}\right)$ in the frequency domain, which is determined by the superposition of the background pressure field $\left(p_{b}\right)$ and the dispersed pressure field (p), expressed as $p_{t}=p_{b}+p$.

Equation (24) is lossless, but may be easily extended to the lossy case. Viscothermal losses of air in the impedance tube are taken into account using the following wave number and impedance expressions [31]:(25)$$k=\frac{\omega }{c}\left(1+\frac{k}{s}\left(1+\left(\gamma -1\right)/\chi \right)\right)$$
(26)$$Z=\frac{\rho_{0}c_{0}}{s}\left(1+\frac{\kappa }{s}\left(1-\left(\gamma -1\right)/\chi \right)\right)$$
being s=R/δ, where R is the inner radius considered in each perforation and $\delta =\sqrt{2\mu /\rho_{0}\omega }$ is the thickness of the viscous boundary layer, μ is the viscosity of air, κ $=\left(1+i\right)/\sqrt{2}$ is the thermal conductivity, γ is the heat capacity of air, and $\chi =\sqrt{P_{r}}$ is the square root of the Prandtl number.

## 4. Materials and Methods

### 4.1. Polyurethane Foam and Textile Fabric Materials

The recycled polyurethane foam (RPU) of thickness t = 2 ± 0.15 cm used is a porous cellular absorbent material and it is made from the residues of the manufacturing processes in the textile industry without chemical treatment inflicting less damage to environment (see Figure 1). These foams present good sound absorbing properties and they are widely used in several applications as absorbent materials.

In our study, samples of RPU will be covered with two different woven textile fabrics with different yarn densities (T1 and T2, see Figure 2) to evaluate the acoustic performance of the coupled system. It should be noted the reduced thickness of these textiles (less than one cm). Both woven textile fabrics T1 and T2 used are flexible woven materials consisting of a network of synthetic fibers made of textured PET polyester increasing the covering capacity. This kind of fabrics possess excellent properties such as low surface yarn density, low production cost, and surface inhomogeneity. The one referenced as T1 is comprised of polyester yarns and a weft yarn of blue chenille with a ratio one chenille/two polyester, which creates a bubble texture increasing the thickness. The other fabric labeled as T2 is comprised of polyester yarns and a yarn of pink chenille.

### 4.2. Drilling Configurations and Methods

The perforated RPU samples were performed in FabLab Océano Naranja (València, Spain) and they were drilled by using two different technologies: a CNC milling machine and a CNC LaserCL2010. CNC is a computer numerical control, which consists of a programmed code with instructions to make precise movements. The CNC machine interacts with a computer equipped with drawing software that transforms the numerical code into Cartesian coordinates. The CNC machines work with a very high degree of precision (tolerance of ± 0.01 mm in every cut) transforming a virtual object (CAD model) into a real one.

RPU foams were perforated with different aperture sizes (D) and different perforation rates ($\emptyset_{M}$) considering a complete perforation depth in all the samples. The three-axis CNC milling machine allows the subtraction of the material when it is drilled (see Figure 3a), while the laser method concentrates light on the surface of the sample and burns it (see Figure 3b).

Figure 4 shows the Field Emission Scanning Electron Microscope (FESEM) images with mod. ZEISS ULTRA55 used to visualize details with high resolution down to 100 µm of the hole surfaces of the perforated RPU foams and to assess the condition of the samples after the drilling process. These foams were tested with suitable accelerating voltage of 2 kV, a working distance up to 10 mm, a 10 µm lens aperture size and 50× magnifications.

Figure 4 also provides information about the effect of the surface roughness of the RPU sample. It can be seen that the CNC laser drilling technology heats the foam material by applying focused laser energy, causing it to vaporize and producing a smooth surface throughout the holes (Figure 4a). The CNC milling technology is based on a subtraction process and it shows higher roughness around the drilled holes than the laser method (see Figure 4b).

Samples have been drilled with different configurations in periodic lattices. All samples are drilled with circular holes of diameter D arranged in square lattice with lattice constant *a*, which corresponds to the distance between two adjacent cells (see Figure 5).

The perforation ratio ($\emptyset_{M}$) is defined as the ratio between volume occupied by the holes and the total volume of the sample,
(27)$$\emptyset_{M}=\frac{n}{4}{\left(\frac{D}{R}\right)}^{2}\cdot 100$$
where n is the number of holes, D is the diameter of the holes and R is the radius of the samples.

In this work, two values of $\emptyset_{M}$ (5% and 20%) and two values of D (4 mm and 6 mm) have been measured, leading to four different configurations as shown in Table 1. The diameter of all samples is 4 cm. A schematic representation (real scale) of the top view of the perforated samples are shown in Figure 6.

### 4.3. Experimental Methodology

A campaign of measurements of both perforated polyurethane foams and textile fabrics based on impedance tubes have been carried out in order to characterize them acoustically as absorbents by using two classical techniques: one based on measuring the sound absorption coefficient at normal incidence as described in [32] and other based on measuring the specific airflow resistance as described in [33]. The samples of RPU and textile fabrics for the normal incidence sound absorption coefficient and the airflow resistivity test were cut circularly with the same impedance tube diameter of ≈ 4 cm. These samples are geometric circular cylinders and both faces are parallel and flat. Also, these samples are fitted easily to the sample holder avoiding an improper adjustment. In the fabric-RPU configuration, the woven fabric is attached to the foam by using sewing needles, whose thickness is less than ≈ 1 mm and do not significantly affect the acoustic measurements.

#### 4.3.1. Sound Absorption Coefficient

The sound absorption coefficient ($\alpha_{n}$) as a function of the frequency was measured by using the transfer function mode described in the Standard ISO 10534-2:1998 [32]. This test method requires an impedance tube device, a data acquisition system (Pulse LabShop v.22.2.0.197), two microphones (½-inch free-field Brüel and Kjӕr pressure microphones − type 4190), a PC and a sound source (Beyma CP800Ti loudspeaker). The sound source generates a plane wave inside the impedance tube hitting the material perpendicularly. The impedance tube is a rigid, methacrylate, smooth, transparent, and airtight duct with circular cross section of 4 cm (see a scheme in Figure 7), which meets the specifications described in the standard [32].

Measurements cover the frequency range from 400 to 3150 Hz. These frequencies are determined by the restrictions imposed by the distance between both microphones, the precision of the signal processing equipment and the diameter of the impedance tube. Each test of the sound absorption coefficient is performed three times for each sample always using the same mounting conditions in order to reduce the dispersion error produced by the inhomogeneity of the samples. The material must fit perfectly to the sample holder without compressing it. The test instrumentation is connected as indicated in Figure 7.

When using the two microphone technique, there is a mismatch between the microphones in the determination of the transfer function. To correct it, the measurement is repeated with the channels exchanged obtaining the transfer function $H_{12}^{\prime }$ and $H_{12}^{\prime \prime }$ in order to calculate the calibration factor using the Equation (28)
(28)Hc=(H12′/H12″)12=|Hc|ejϕc
where $\phi_{c}$ is the corrected phase angle.

From a MATLAB function implemented for this case, the corrected transfer function, $H_{12}$, which describes the total acoustic field, can be obtained as
(29)$$H_{12}=\frac{{\hat{H}}_{12}\;}{H_{c}}=\frac{e^{{\mathrm{jk}}_{0}x_{2}}+{\mathrm{re}}^{-{\mathrm{jk}}_{0}x_{2}}}{e^{{\mathrm{jk}}_{0}x_{1}}+{\mathrm{re}}^{-{\mathrm{jk}}_{1}}}$$

The complex acoustic transfer function ($H_{12}$) of the signals in both microphones is determined in order to obtain the complex acoustic pressure reflection coefficient for a plane wave at normal incidence (r). It represents the complex ratio of the pressure amplitude of the reflected wave and the incident wave, as
(30)$$r=\frac{H_{12}-H_{I}}{H_{R}-H_{12}}e^{2{\mathrm{jk}}_{0}x_{1}}$$
where $H_{I}=e^{\mathrm{jks}}$ is the sound pressure transfer function of the incident wave, $H_{R}=e^{-\mathrm{jks}}$ is the sound pressure transfer function of the reflected wave, $k_{0}=2\pi f/c_{0}$ is the wave number being f the frequency and $x_{1}$ is the distance between the sample and the microphone placed further away from it.

After the complex reflection coefficients are obtained, the normal incidence sound absorption coefficients follow from Equation (15).

#### 4.3.2. Airflow Resistivity

The airflow resistivity (σ) is defined as the airflow resistance divided by the sample thickness t. It is directly related to the absorption properties of the porous materials. It is measured under certain limitations of our experimental setup considering the indirect method described by Ingard and Dear (1985) [33]. In this indirect model, the measuring device is a rigid methacrylate tube of circular cross section with a sound source at one end; in the other, a completely rigid termination. The Mic.1 is located in front of the sample in order to directly measure the sound pressure (P_1_) and the Mic.2 is set at the end of the impedance tube, near the rigid termination (P_2_). In Figure 8, a schematic of the experimental device is shown, where the placement of the microphones and the sample can be seen.

The cross section of the impedance tube must be very small compared to the wavelength of the sound (λ) to ensure plane wave propagation. Therefore, the following condition: λ ≫ 1.7D, where D is the inner diameter of the tube, is satisfied. It is possible to calculate the airflow resistance by finding the minimum of the imaginary part of the pressure ratio $p_{1}/p_{2}$ for frequencies that satisfy the following condition: L + t = (2n − 1)λ/4, where n is an integer number.

The average values of the specific airflow resistivity are obtained by using the absolute value of the imaginary part of the transfer function between the signals of the two microphones as
(31)σ≈(ρ0c0t)|Im(1H12)|

## 5. Results and Discussion

### 5.1. Effect of Perforation Rate and Aperture Size On the Sound Absorption

There are different technologies for material drilling depending on their hardness, their composition and their manufacture. In the case of polyurethane foams, these can be cut efficiently using two main techniques: laser and drilling machine. When using laser method, the laser beam interacts with a material causing its vaporization on its way for a given depth. Through the drilling method, the perforation is not perfect because the perforated hole, after removing the drill, is refilled with fibers [34].

The effects of drilling operation on the surface roughness depend on the perforation technique employed. Milling is a traditional process for hole-making by using rotary cutters to remove material. The quality of the drilled holes usually is improved by secondary drilling. Laser cutting is one of the most extended methods for industrial applications [35].

The effect on the sound absorption in samples perforated with two different drilling methods has been tested. It has been checked that both configurations have a very similar acoustic behavior. Consequently, in this work, RPU foams perforated only with the milling machine technique are analyzed.

A study of samples with D = 4 mm and different $\emptyset_{M}$ has been carried out. Figure 9 shows the results obtained for the sound absorption coefficient of the RPU samples placed at the end of the impedance tube. At mid frequencies up to 2000 Hz, the unperforated sample has a sound absorption coefficient higher than the perforated samples. It can be seen in Figure 9 that the maximum of absorption shifts towards higher frequencies when the perforation rate is increased. In this type of perforated configurations, both the maximum sound absorption value and the frequency where the absorption is maximum, varies depending on the rate and the diameter of perforation chosen [19]. The absorption values in the region 2000 to 3150 Hz are the maximum values in the frequency range that can be observed due to the configuration of the measurements. Examples of this acoustic behavior can be observed in [9]. The $\emptyset_{M}=\;$5% configuration has the maximum absorption at the frequency of 2500 Hz, and the $\emptyset_{M}=\;$20% at the frequency of 3150 Hz.

### 5.2. Effect of the Textile Fabric over a RPU Foam on Acoustic Absorption

A textile fabric has been added over a RPU foam in order to improve the sound absorption. In Table 2, the woven fabric characteristics of both fabrics T1 and T2 are shown. T2 fabric has higher surface yarn density and higher airflow resistivity. Figure 10 shows the study of the normal incidence sound absorption coefficient of both textile fabrics. The sound absorption of both fabrics is very low at 400 Hz showing a slight increase with frequency, being smaller than 0.3 in the whole frequency range.

Figure 11 shows RPU samples with two different textile fabrics. The normal incidence sound absorption coefficient of the perforated RPU foam in combination with the woven fabrics for D = 4 mm is measured and shown in Figure 12. The acoustic influence of using textiles that cover RPU foams is analyzed.

It is intended to achieve a configuration that provides an increase in the airflow resistance of the sample set as this acoustic parameter can be considered additive in this union of materials. For sufficiently low-density materials, if the airflow resistance of the fabric-RPU sample is increased, an increase of the acoustic impedance is generated increasing the sound absorption (see Figure 12). Examples of this phenomenon can be seen in reference [36], where the acoustic absorption of the fabric + air configuration is much greater than the absorption of the fabric. This is achieved by increasing the impedance of the material set compared to the impedance of the air cavity. It can be seen in Figure 12 that the effect of using a textile fabric to cover the RPU sample results in a decrease of the absorption peak. The additional mass introduced by the fabric, explains why the resonance frequency of the RPU + fabric configuration is reduced. In both figures (see Figure 12a,b), the same trend is observed: the absorption coefficient is enhanced by placing a fabric on the surface of both unperforated and perforated RPU samples. An increase in the sound absorption coefficient is produced at mid frequencies with the combination textile fabric + RPU compared to the foam without fabric. The highest sound absorption coefficient is reached at 1600 Hz when RPU is combined with the T2 fabric. The presence of the fabric in the RPU foam results in a displacement of the sound absorption maximum towards low frequencies. In Figure 12a, the combination T1 fabric + perforated RPU foam presents higher absorption than the combination T1 fabric + unperforated RPU at high frequencies, from 1350 to 3150 Hz. Figure 12b evidences greater difference in absorption when perforation is considered. In both cases, the sound absorption coefficient reaches quasi perfect absorption at the maximum.

### 5.3. Comparison of Predicted and Measured Values

The sound absorption coefficient in normal incidence for a set of four different configurations of perforated RPU samples was experimentally measured. In order to validate the theoretical and numerical models proposed in Section 2 for double porosity media, predicted and measured sound absorption curves have been compared. As it can be seen in Figure 13, there are no significant differences between the models and the measurements. A similar matching has been observed in the other configurations studied, although they are not presented here for simplicity. Both models used the acoustic parameters of both unperforated and perforated RPU foams, with a thickness of 2 ± 0.15 cm, shown in Table 3 in order to describe the sound absorption properties of perforated foams. There are slight deviations at mid frequencies that can be explained from the experimental uncertainty.

Two different configurations of perforated RPU foams covered by two different textile fabrics have been measured experimentally. Figure 14a–d compare the predicted and measured absorption coefficients of the absorbent structure in each configuration. It can be seen in all figures a good agreement between the measurements and both models, analytical and numerical, that have been previously presented in Section 2. In the numerical model, the values of the Delany and Bazley coefficients are presented in Table 4 for both textile fabrics (T1 and T2). It is important to remark here that the model for viscothermal losses in the air perforations has not been considered in numerical simulations. This model is valid for open perforations and here the presence of the fabric avoids radiation. The slight difference between simulated and experimental sound absorption values observed in all cases of Figure 14 may be due to the composition of the fabric used and the imperfections between the porous material and the textile.

## 6. Conclusions

The influence of adding fabric to the surface of recycled polyurethane perforated foam samples has been investigated. The normal incident sound absorption coefficient of samples were measured experimentally in an impedance tube. The influence of perforation parameters like the perforation rate and the aperture size have been investigated. The higher the perforation rate, the lower the sound absorption values at medium and high frequencies. For the same perforation rate, the larger the aperture size, the absorption values are slightly higher. The combination textile fabric + perforated RPU foam modifies and increases significantly the sound absorption coefficient at frequencies around 2 kHz compared to unperforated RPU samples. Experimental results evidence that the sound absorption depends slightly on the type of the textile fabric used. A theoretical model and a numerical model was performed to validate the results presented in this work. Good agreement is found between both models and measurements.

## Figures and Tables

**Figure 1 polymers-12-00401-f001:**
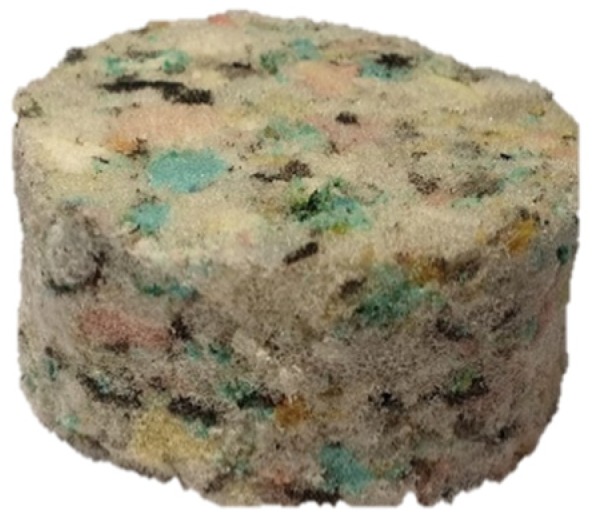
Recycled polyurethane foam used in this work.

**Figure 2 polymers-12-00401-f002:**
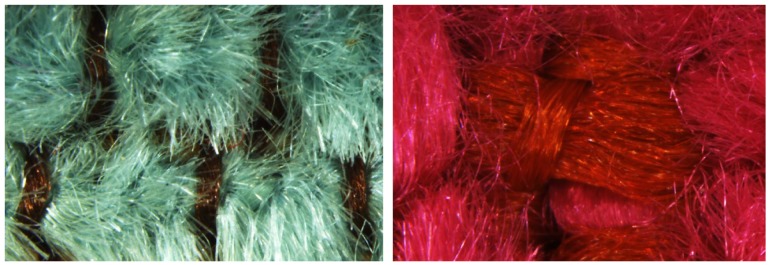
Surface and the warp/weft pathways of the textile materials is analyzed with microscope imaging technique with 16X magnifications. **left**: T1; **right**: T2.

**Figure 3 polymers-12-00401-f003:**
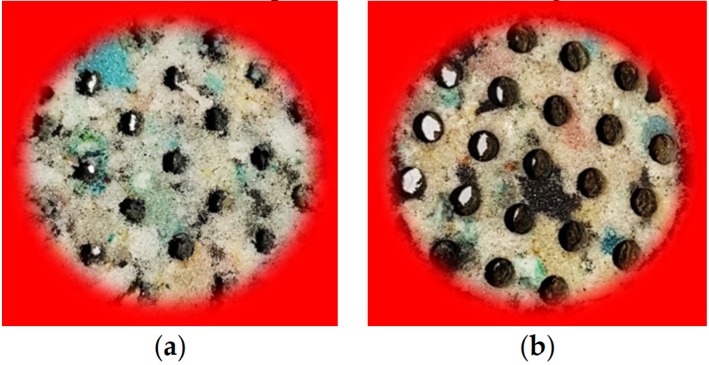
Specification of drilled RPU foams: D = 6 mm and $\emptyset_{M}$ = 20%. (**a**) milling machine method; (**b**) laser method.

**Figure 4 polymers-12-00401-f004:**
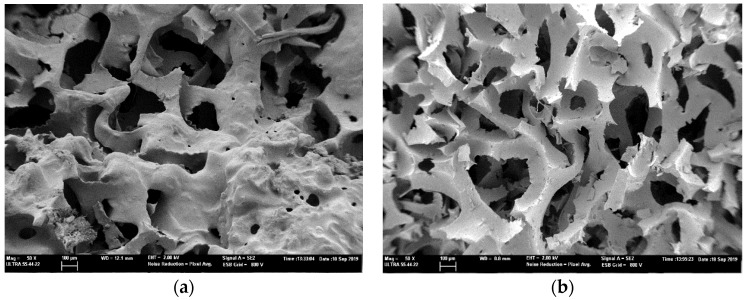
Electron microscopy images of surface of the perforated foam sample with (**a**) laser and (**b**) milling machine.

**Figure 5 polymers-12-00401-f005:**
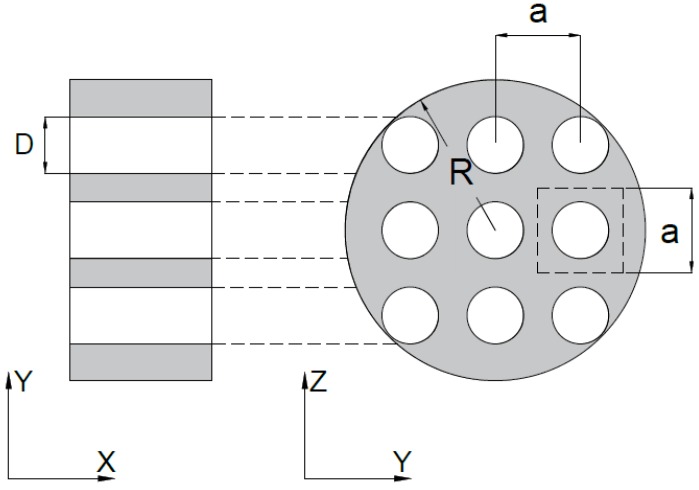
General configuration of 2D periodic perforated samples arranged in a square lattice with lattice constant a. D is denoted as the diameter of the perforated hole and R is the radius of the foam sample.

**Figure 6 polymers-12-00401-f006:**
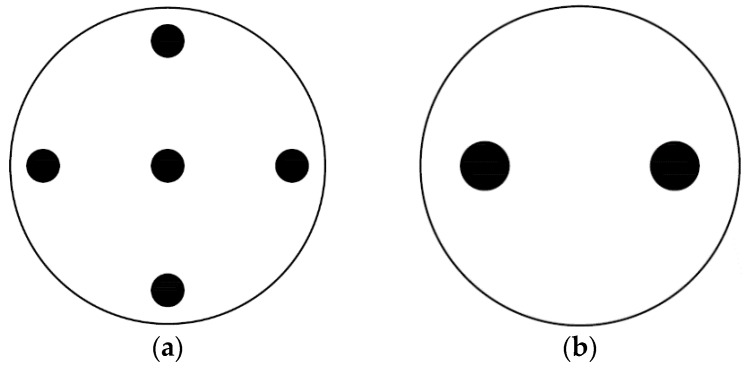
Four different drilling configurations have been measured: (**a**) D = 4 mm, $\emptyset_{M}$ = 5%, a = 15.8 mm; (**b**) D = 6 mm, $\emptyset_{M}$ = 5%, a = 23.8 mm; (**c**) D = 4 mm, $\emptyset_{M}$ = 20%, a = 7.9 mm; and (**d**) D = 6 mm, $\emptyset_{M}$ = 20%, a = 11.9 mm. The radius of all samples is R = 2 cm.

**Figure 7 polymers-12-00401-f007:**
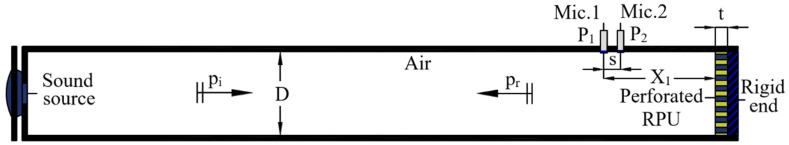
Diagram of the impedance tube used to measure the sound absorption at normal incidence. D is the inner diameter of the tube (D = 4 cm); t is the thickness of the sample, X_1_ is the distance between Mic.1 and the sample; p_i_ is the acoustic pressure of the incident wave; p_r_ is the acoustic pressure of the reflected wave; and s is the separation between both microphones (s = 3.2 cm).

**Figure 8 polymers-12-00401-f008:**
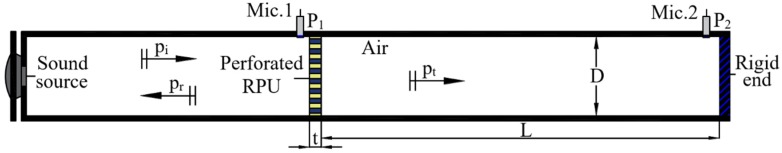
Schematic diagram of the impedance tube used to measure specific airflow resistance by Ingard and Dear method. L is the distance between the back face of the sample and the rigid end; t is the thickness of the sample; p_t_ is the acoustic pressure of the transmitted wave.

**Figure 9 polymers-12-00401-f009:**
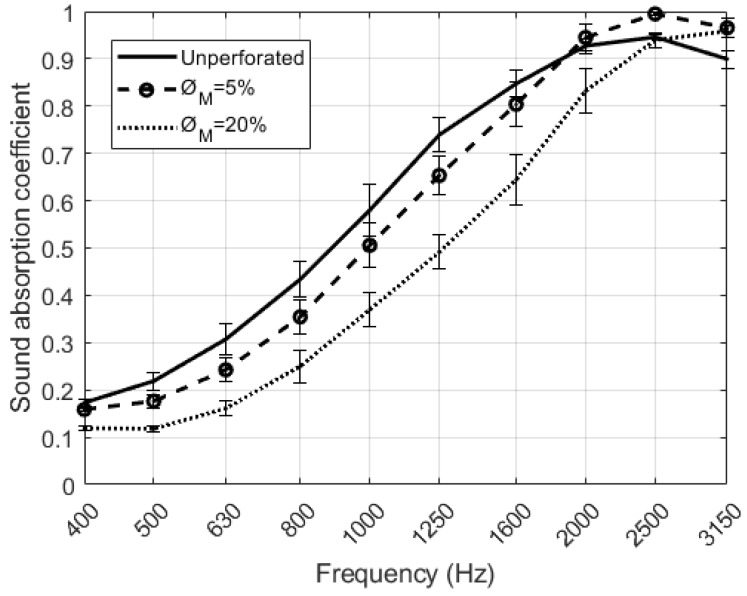
Results of the normal incidence sound absorption coefficient of the perforated RPU foam samples for D = 4 mm.

**Figure 10 polymers-12-00401-f010:**
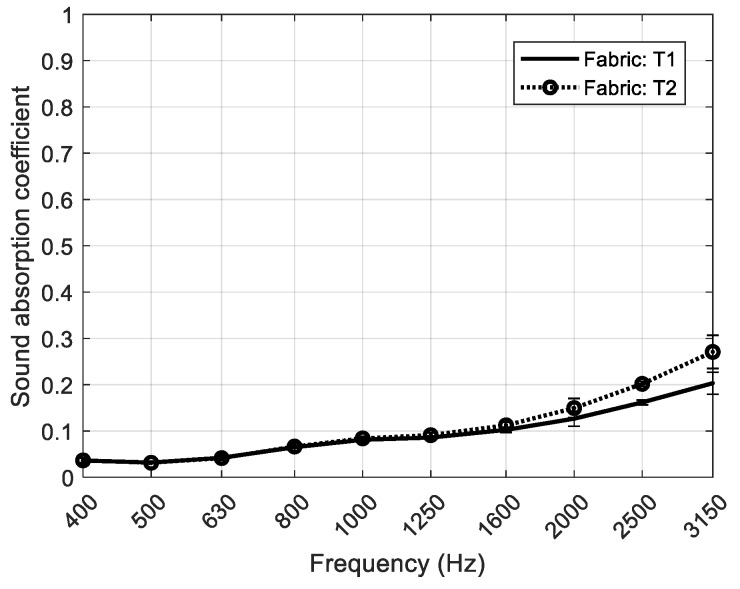
Sound absorption coefficient at normal incidence of the textile fabrics with their dispersion percentage.

**Figure 11 polymers-12-00401-f011:**
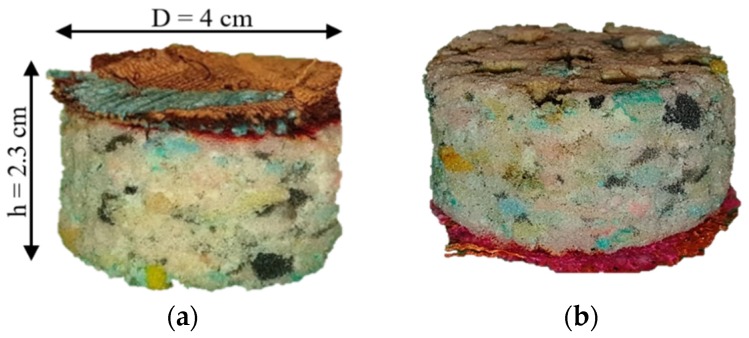
Two different configurations of the system perforated fabric−RPU. (**a**) T1−RPU; (**b**) T2−RPU.

**Figure 12 polymers-12-00401-f012:**
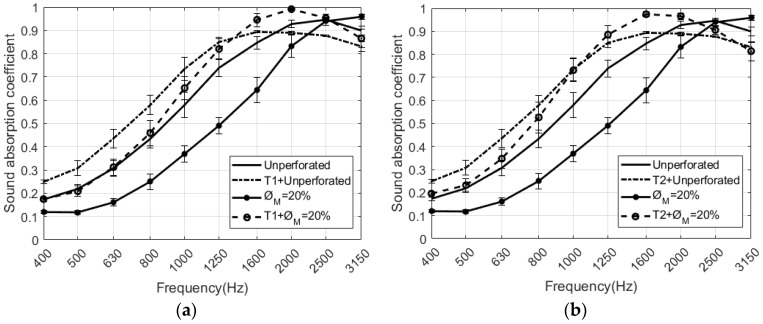
Results of the sound absorption coefficient at normal incidence of the different combinations between unperforated and perforated RPU samples and textile fabrics (T1 and T2): (**a**) $\emptyset_{M}$ = 20%, D = 4 mm, T1; (**b**) $\emptyset_{M}$ = 20%, D = 4 mm, T2.

**Figure 13 polymers-12-00401-f013:**
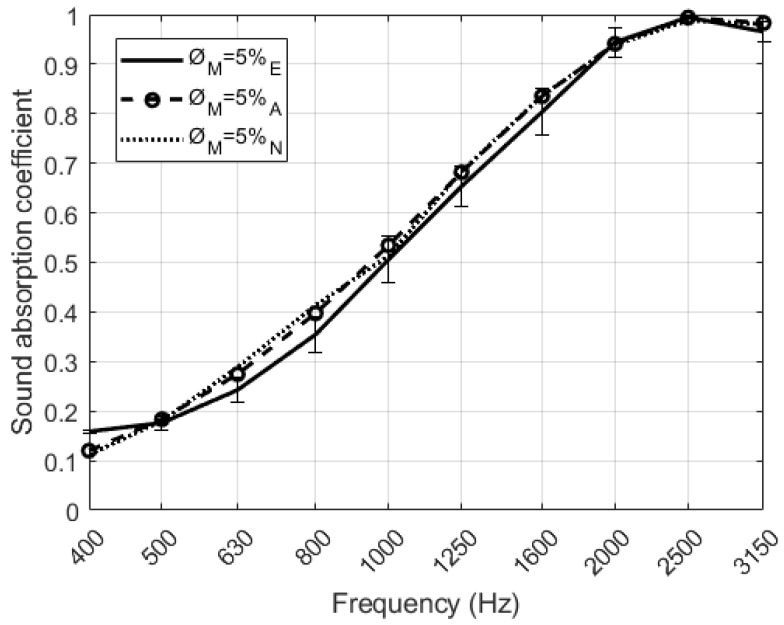
Normal incidence plane wave sound absorption coefficient of the RPU samples for D = 4 mm. Comparison between measurement (**continuous lines**) theoretical model (**dashed lines**) and numerical model (**dotted lines**).

**Figure 14 polymers-12-00401-f014:**
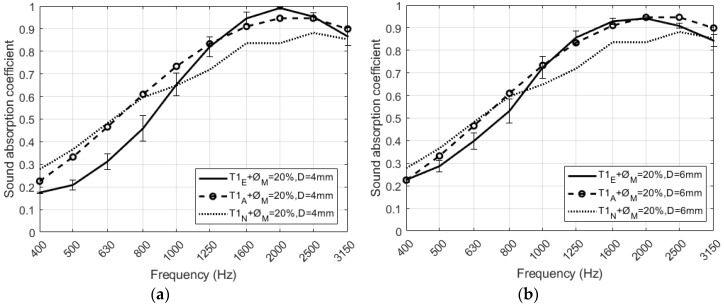
Normal incidence plane wave sound absorption coefficient of the RPU samples covered with two different textile fabrics. Comparison between measurement (**continuous lines**) theoretical model (**dashed lines**) and numerical model (dotted lines) of (**a**) T1 fabric with 20% and D = 4 mm, (**b**) T1 fabric with 20% and D = 6 mm, (**c**) T2 fabric with 20% and D = 4 mm, (**d**) T2 fabric with 20% and D = 6 mm.

**Table 1 polymers-12-00401-t001:** Detailed parameters of the four drilling configurations measured

Configuration	${Ø}_{M}\;\mathrm{real}\;(\%)$	${Ø}_{M}\;\mathrm{approx}.\;(\%)$	D (mm)	*a* (mm)	*No. of holes*
a)	5	5	4	15.8	5
b)	4.5	5	6	23.8	2
c)	21	20	4	7.9	21
d)	20.25	20	6	11.9	9

**Table 2 polymers-12-00401-t002:** Thickness (t), surface yarn density (ρ), and airflow resistivity (σ) of the woven textile fabrics

Fabric Type	t (cm)	ρ $\;(\mathrm{Kg}\cdot 10^{-3}\cdot m^{-2})$	$\sigma (\mathrm{kPa}\cdot s\cdot m^{-2})$
T1	0.18	398	204−250
T2	0.14	478	270−274

**Table 3 polymers-12-00401-t003:** Acoustic parameters of the samples measured. Bulk density of the material ($\rho_{m}$), density of the fibers ($\rho_{f}$), airflow resistivity (σ), porosity (ø), meso-porosity (${ø}_{M}$), tortuosity ($\alpha_{\infty }$), viscous characteristics length (Λ), thermal characteristics length ($\Lambda^{\prime }$), and mean fiber diameter (d).

Foam Type	$\rho_{m}$ $\left(\mathrm{Kg}\cdot m^{-3}\right)$	$\rho_{f}$ $\left(\mathrm{Kg}\cdot m^{-3}\right)$	σ $\left(\mathrm{kPa}\cdot s\cdot m^{-2}\right)$	∅	${Ø}_{m}$	$\alpha_{\infty }$	Λ (µm)	$\Lambda^{\prime }$(µm)	d (µm)
Unperforated	182	1250	20.4−20.5	0.88	0	1.07	52.1	104.2	25
5%4 mm	153				0.05				
5%6 mm	165				0.05				
20%4 mm	130				0.2				
20%6 mm	132				0.2				

**Table 4 polymers-12-00401-t004:** Delany and Bazley coefficients of fabrics T1 and T2.

Fabrics	$C_{1}$	$C_{2}$	$C_{3}$	$C_{4}$	$C_{5}$	$C_{6}$	$C_{7}$	$C_{8}$
T1	0.1101	0.6283	0.1846	0.7027	0.0455	0.8173	0.0733	0.7487
T2	0.0984	0.7225	0.2191	0.6716	0.0431	1.0000	0.0899	0.2333

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
