# Peer review of "Sound Absorption Properties of Perforated Recycled Polyurethane Foams Reinforced with Woven Fabric"

_polymers, 2020, doi:10.3390/polym12020401_

Round 1

Reviewer 1 Report

Please provide the more information about what type of RPU and textile fabrics were used in the study. Add it in the section 4. Please add the procedure about how the RPU and fabrics composites was prepared. The figure 10 shows that fabrics have lower sound absorption coefficient than RPU. But the authors have used it to improve the sound absorption of RPU. Please provide the reason for it in the results and discussion section. Please provide some hypothesis for the higher sound absorption coefficient of perforated foam in the region 2000 to 3150 Hz (Figure 9).

Reviewer 2 Report

 The sound absorption by polyurethane foams  combined with fabrics was investigated, which deserves to be accepted after some minor revisions:
(1) The units of every physical quantity have to be given in all Eqs.;
(2) "analysed" or "analyzed"?
(3) Besides the measured macropore sizes, the sizes of mesoporous pores are suggested to be provided;
(4) More discussion should be given to the absorption mechanisms.
